# MAARS: Multiagent Actor–Critic Approach for Resource Allocation and Network Slicing in Multiaccess Edge Computing

**DOI:** 10.3390/s24237760

**Published:** 2024-12-04

**Authors:** Ducsun Lim, Inwhee Joe

**Affiliations:** Department of Computer Software, Hanyang University, Seoul 04763, Republic of Korea; imcoms@hanyang.ac.kr

**Keywords:** multiaccess edge computing, reinforcement learning, network slicing, resource allocation

## Abstract

This paper presents a novel algorithm to address resource allocation and network-slicing challenges in multiaccess edge computing (MEC) networks. Network slicing divides a physical network into virtual slices, each tailored to efficiently allocate resources and meet diverse service requirements. To maximize the completion rate of user-computing tasks within these slices, the problem is decomposed into two subproblems: efficient core-to-edge slicing (ECS) and autonomous resource slicing (ARS). ECS facilitates collaborative resource distribution through cooperation among edge servers, while ARS dynamically manages resources based on real-time network conditions. The proposed solution, a multiagent actor–critic resource scheduling (MAARS) algorithm, employs a reinforcement learning framework. Specifically, MAARS utilizes a multiagent deep deterministic policy gradient (MADDPG) for efficient resource distribution in ECS and a soft actor–critic (SAC) technique for robust real-time resource management in ARS. Simulation results demonstrate that MAARS outperforms benchmark algorithms, including heuristic-based, DQN-based, and A2C-based methods, in terms of task completion rates, resource utilization, and convergence speed. Thus, this study offers a scalable and efficient framework for resource optimization and network slicing in MEC networks, providing practical benefits for real-world deployments and setting a new performance benchmark in dynamic environments.

## 1. Introduction

Advancements in geospatial remote sensing technologies, driven by increased spectral, spatial, and temporal resolutions [1,2], have revolutionized various fields, including national economies, ecological protection, and national security. These technologies enable precise observations and analyses for diverse applications, including Earth observation [3] and the Internet of Things (IoT) [4], significantly improving the accuracy of information collection and analysis. Real-time processing of high-resolution data is now crucial for applications such as climate change monitoring, natural disaster response, and infrastructure management. This need for real-time analysis aligns perfectly with the capabilities of multi-access edge computing (MEC), a transformative approach to handling time-sensitive data.

Recent breakthroughs in artificial intelligence, edge computing, and the IoT have enabled more sophisticated analyses of geographic phenomena, earth science processes, and intelligent decision-making. In particular, MEC [5] has emerged as a key technology, enhancing real-time data processing and response times by bringing computation and storage closer to the network edge. This shift has facilitated faster analysis of remote sensing data, crucial for the sustainable development of nations and cities [6,7,8]. However, efficient resource allocation in MEC networks, especially with high traffic and dynamic user demands, remains a significant challenge.

MEC networks provide highly efficient, real-time computing services for smart device users (SDUs). By processing tasks locally at edge nodes (ENs) instead of relying on centralized clouds, MEC minimizes latency and conserves bandwidth [9], offering significant advantages in environments with large-scale IoT deployments, such as smart cities, autonomous vehicles, and industrial automation systems.

However, a major challenge in MEC architectures is the efficient allocation and management of the limited network resources. Network-slicing (NS) techniques address this challenge by logically partitioning the network and allocating resources based on service requirements, ensuring ultralow latency and high reliability. NS effectively virtualizes physical infrastructure, enabling flexible adaptation to diverse user needs [10].

Beyond computational resources, spectrum and power allocation are critical aspects of resource management in MEC networks. Effective power allocation optimizes energy efficiency and mitigates interference, a particularly complex task in high-traffic environments. Additionally, spectrum allocation ensures efficient use of available bandwidth to support diverse user requirements. Recent research, such as that by Ngene et al. [11], highlighted the importance of these factors, especially power allocation strategies, in resource management frameworks for next-generation communication systems.

Reinforcement learning (RL)-based resource allocation and NS algorithms have attracted considerable attention for their ability to dynamically adjust resource allocation in response to network-state changes, effectively handling unpredictable traffic patterns and user demands [12]. Various RL algorithms, such as Q-learning [13], deep Q-network (DQN) [14], policy gradient [15], and proximal policy optimization (PPO) [16], have been actively studied to optimize network-resource management.

This study proposes a novel approach to address NS and resource allocation challenges in MEC networks, aiming to maximize the success rate of user-computing tasks in each network slice through real-time monitoring and efficient network-resource allocation. Specifically, these problems are decomposed into two subproblems, i.e., efficient core-to-edge slicing (ECS) and autonomous resource slicing (ARS), and a multiagent actor–critic resource scheduling (MAARS) algorithm is proposed to handle them. MAARS combines MADDPG for ECS and SAC for ARS to optimize resource distribution and real-time resource management, respectively.

This integration of MADDPG and SAC creates a unique two-tier optimization framework tailored for MEC networks. MADDPG effectively manages multiagent coordination in distributed environments, enabling precise core-to-edge resource allocation for ECS. SAC, employed for ARS, provides stable and efficient real-time decision-making in scenarios with continuous action spaces. The synergistic combination of these algorithms ensures seamless resource management across the entire network.

To further enhance the performance of MADDPG, we introduce model-agnostic metalearning (MAML) for metalearning initialization. MAML pre-trains the MADDPG neural networks with optimized meta-policy parameters, enabling rapid adaptation to dynamic network conditions. This approach reduces training iterations and improves efficiency and scalability, ensuring robust performance under fluctuating conditions.

This innovation is critical for addressing the challenges in MEC networks, where rapid adaptation is essential. MAML equips MADDPG to handle diverse resource demands effectively, ensuring lower latency, better resource utilization, and higher task completion rates. Compared to traditional methods, this approach significantly reduces convergence time and improves the adaptability of resource allocation strategies.

Extensive experiments demonstrates the superiority of the proposed approach. The MAARS framework outperforms existing state-of-the-art methods, achieving higher user task completion success rates, improved resource utilization, and reduced convergence time. This is attributed to the integrated MADDPG and SAC framework, enabling independent yet interconnected optimization of ECS and ARS tasks.

By effectively decomposing complex resource allocation problems and applying advanced multiagent RL techniques, MAARS significantly advances MEC network management. The innovative integration of MADDPG, SAC, and MAML enhances computational efficiency and ensures robust and dynamic adaptation to evolving network conditions, setting a new benchmark for MEC NS and resource allocation.

The remainder of this paper is organized as follows. Section 2 reviews the related research in this field, and Section 3 presents the system model and mathematical formulation of the problem. Section 4 describes the proposed algorithm and experimental conditions, Section 5 presents the simulation results for evaluating the algorithm performance, and Section 6 discusses them. Finally, Section 7 concludes the study and presents directions for future research.

## 2. Related Work

NS and resource allocation in MEC environments have been extensively studied [17,18,19,20,21,22,23,24,25,26,27,28]. Researchers have explored various approaches to optimize these processes. For instance, Tran et al. [17] proposed a mixed-integer nonlinear programming approach with heuristic algorithms to optimize task offloading and resource allocation in MEC networks, improving offloading utility under resource constraints. However, it faces challenges in adapting to dynamic network conditions in real-time. To overcome this limitation, Cevallos et al. [18] introduced a dueling double deep Q-network (D3QN) for 5G content delivery networks, enhancing adaptability to network complexity and reducing operational costs.

Deep reinforcement learning (DRL) has also emerged as a promising solution. Huang et al. [19] developed a DRL-based offloading algorithm to minimize energy consumption in mobile devices while meeting real-time task deadlines; however, scalability remained a concern when handling numerous simultaneous tasks. Similarly, Wang et al. [20] used DRL for dynamic resource allocation and service time reduction, but encountered challenges associated with learning speed and stability. Nduwayezu et al. [21] addressed subcarrier allocation in non-orthogonal multiple access (NOMA) MEC systems using a DRL-based task offloading algorithm, achieving significant improvements in resource utilization and computation speed. Ning et al. [22] further extended this work by focusing on resource allocation in multiuser environments, enabling dynamic MEC server–user interactions to reduce computation costs.

Slicing frameworks have proven crucial for enhancing network scalability and performance. Shah et al. [23] and Khan et al. [24] developed frameworks for scalable dedicated networks and dynamic resource adjustments based on traffic criticality. Wu et al. [25] proposed a two-tier constraint algorithm for resource allocation in vehicular-communication systems to ensure reliable services in high-traffic conditions. Seah et al. [26] introduced a latency-aware resource scheduler to minimize latency while maintaining system fairness. Jin et al. [27] and Chiang et al. [28] applied deep RL techniques to resource slicing, reducing latency and enhancing computational efficiency in MEC systems.

Building upon these advancements, this paper introduces the MAARS algorithm, a novel two-stage slicing model designed to address NS and resource-allocation challenges in MEC environments. MAARS employs multiagent RL techniques—MADDPG for ECS and SAC for ARS—to enable dynamic and collaborative resource management. Additionally, by integrating MAML for neural network initialization, MAARS accelerates learning and enhances adaptability to network changes, thereby improving task success rates and overall system performance.

This study contributes to the field by proposing an innovative two-stage slicing model that effectively addresses NS and resource-allocation issues in MEC networks. The model combines ECS and ARS to efficiently allocate network resources while adapting to real-time demands. By leveraging RL for dynamic resource allocation, the proposed framework offers better flexibility than existing methods, maximizing network performance and significantly increasing the success rates of user-computing tasks.

## 3. System Model

This section describes the MEC-based network architecture, and the proposed system model based on this architecture for handling tasks.

### 3.1. System Architecture

Figure 1 illustrates an MEC network comprising numerous ENs and SDUs. SDUs generate different types of computing tasks to fulfill the requirements of various services, such as augmented reality, virtual reality, video streaming, and online gaming. Each EN comprises a base station (BS) and an associated EN, which provide network bandwidth and computing resources, respectively. Network function virtualization (NFV) abstracts the network infrastructure into virtual resources to enable flexible resource management, whereas software-defined networking (SDN) flexibly configures network functions to provide optimal network performance.

The MEC network provides logical access to all network-based SDUs. Each network has a set of BSs and a corresponding set of associated ENs. Each BS has a certain amount of network bandwidth, and each EN provides a certain amount of computing resources. The network resources comprise the bandwidth and computing resources of all BSs and ENs. Users generate W computing jobs with different service-delay requirements, which are typically offloaded to the nearest EN based on the location of the user. Each computing task can be offloaded to only one EN, and all ENs in the network exchange information to ensure optimal resource allocation.

This system model leverages NFV and SDN technologies to provide high-quality services to users through dynamic resource allocation and optimization. This increases network flexibility and scalability and maximizes MEC network performance by efficiently handling various service requirements. The notations employed in the mathematical representation of the system model are listed in Table 1.

### 3.2. NS Model

At the network level, the computing and data resources are dynamically allocated according to the real-time service demand of each logical access network. To achieve this, we introduce the ECS and ARS slicing models. The ECS manages resources through cooperation between the central management system of the network and ENs and is responsible for optimization and resource distribution within the network. By contrast, the ARS allocates resources based on the real-time service demands of ENs that are physically close to users and aims to minimize latency. This model enables efficient network-resource management by the central management system and edge-computing nodes, as resources are dynamically allocated based on user demand, thereby optimizing the performance and quality of network services. The resource-allocation set for each logical access network is defined as follows:(1)$$R_{m}\left(t\right)=\{B_{m}\left(t\right),F_{m}\left(t\right),\;C_{m}\left(t\right)\},$$
where $B_{m}\left(t\right)$ denotes the bandwidth allocated to the network at time *t*, $F_{m}\left(t\right)$ denotes the allocated computing resources, and $C_{m}\left(t\right)$ denotes the caching resources. This resource-allocation set illustrates the distribution of network bandwidth, computing resources, and caching resources over time. It represents an efficient resource-allocation process aimed at handling real-time changes in user demands. State information is dynamically collected from each EN, and a virtual controller is assumed to manage this information. Network resources are composed of individual network-resource entities, and the set of these entities is expressed as
(2)$$X_{m}=\left\{X_{m}|\;m=1,\;2,\;\ldots ,M\right\},$$
where X is the specific resource entity assigned to the *m*-th logical network. Each network-resource entity is periodically partitioned based on the current demands of user-computing tasks such that resources are efficiently allocated for different user tasks. This process is expressed as follows:(3)$$Z_{w}\left(t\right)=\{B_{w}\left(t\right),\;F_{w}\left(t\right),\;C_{w}\left(t\right)\},$$
where $B_{w}\left(t\right)$, $F_{w}\left(t\right)$, and $C_{w}(t)$ denote the bandwidth required for the w-th service, the computing resources required for it, and the caching resources, respectively. Equation (3) indicates that network resources are dynamically managed over time and appropriately allocated based on real-time changes in user demands. The network-resource entities are efficiently managed through a central management system and virtual controller, thereby enhancing network performance and stability. This is defined as follows:(4)$$X_{m}=\left\{X_{w,\;m}|\;w=1,\;2,\;\ldots ,W\right\},$$
where $X_{m}$ represents the m-th service-resource instance generated by the m-th network-resource entity. These resource entities are dynamically adjusted according to network demands, and network resources are efficiently allocated based on the predictive models and real-time data. This enables continuous monitoring of network performance and allows real-time adjustments to resource-allocation strategies to maintain optimal network performance. For example, when the demand for a specific service increases, the necessary resources are immediately allocated to maintain service quality. Conversely, when demand decreases, resources are reclaimed and reallocated to other services, thereby maximizing the overall efficiency of network resource utilization.

### 3.3. Resource Allocation in Network-Resource Entities

During resource allocation, the computational tasks of each service-resource entity, $Z_{w}$, are dynamically offloaded to the most suitable nearby BS. Specifically, when a user generates a task $u_{w,\;k}\left(t\right)$, it is sent to the nearest BS that can handle it efficiently. At this point, the network continuously monitors the status of computing resources in real-time to ensure optimal resource allocation. This dynamic offloading mechanism enables efficient resource management, thereby ensuring both task completion and network stability.

The BS allocates network-resource units (NRUs) to handle computing tasks and transmits the task data to servers connected to the BS. An NRU comprises the bandwidth, network links, and other network-related resources optimally distributed by the resource-management system. The server then allocates a computing resource unit (CRU) to handle the task. A CRU comprises computing resources, such as a central processing unit (CPU), graphics processing unit (GPU), or memory, and the resource-management system of the server allocates the appropriate resources to meet the task requirements.

The server utilizes its allocated CRUs to process computing tasks and sends the results back to the user. During this process, resource usage is constantly monitored, and resources are redistributed as required to maximize system efficiency. This resource-management process ensures that network and computing resources are maintained under optimal conditions, and users experience low latency and are provided with highly reliable services.

### 3.4. Delay Analysis in Network-Resource Management

The computational tasks of the user are dynamically offloaded to the network-slice segment (NSS) and reconfigured to optimize overall system performance. The *k*-th computing task of a user is represented as a tuple, $u_{w,\;k}\left(t\right)=\left(V_{k}\left(t\right),\;P_{k}\left(t\right),\;{resp}_{w,max}\right)$, where $v_{k}\left(t\right)$, $P_{k}\left(t\right)$, and ${resp}_{w,max}$ denote the data volume, computational complexity, and maximum allowable response time, respectively. Note that defining the performance requirements of each task is essential for precise resource allocation.

The server-selection indicator, ${Svr}_{k,m}\left(t\right)\in \{0,1\}$, determines the MEC server to which a specific task $\phi_{wk}\left(t\right)$ is offloaded. $S_{km}\left(t\right)=1$ indicates that the task is offloaded to a particular server, whereas $S_{km}\left(t\right)=0$ indicates that it is not. To optimize performance, the computational tasks of each user are offloaded to a single MEC server, thus satisfying the following constraint:(5)$$\sum_{m=1}^{M}{Svr}_{k,m}(t)\leq 1.$$

This ensures each task is assigned to only one MEC server, thereby preventing redundant resource allocation and promoting efficient resource usage. The number of network resources allocated to task $\phi_{wk}\left(t\right)$ is denoted as $B_{k.w,m,n}\left(t\right)$, and the transmission rate for the task is defined according to Shannon’s formula as follows:(6)$$R_{k}\left(t\right)=\sum_{n=1}^{N}\left[S_{km(t)}\cdot B_{k.w,m,n}(t)\cdot \log_{2}\left(1+\frac{p_{k}(t)h_{k,m}(t)}{N_{0}+I_{km}}\right)\right],$$
where $p_{k}(t)$ represents the transmission power, $h_{k,m}(t)$ denotes the channel gain, $N_{0}$ is the noise power, and $I_{km}$ denotes interference based on the network-resource state. This equation plays a key role in optimizing the transmission rate for efficient network-resource management. The transmission delay $\Gamma_{k,\;trans}\left(t\right)$ is defined as a function of the data size to be transmitted and the transmission rate, as follows:(7)$$\Gamma_{k,\;trans}\left(t\right)=\frac{V_{k}(t)}{r_{k}(t)},$$
where $V_{k}(t)$ represents the data size and $r_{k}(t)$ denotes the transmission rate. Equation (7) indicates that a lower transmission delay implies higher performance. Next, the processing delay $\Gamma_{k,proc}\left(t\right)$ can be defined based on the task complexity and allocated computing resources as follows:(8)$$\Gamma_{k,proc}\left(t\right)=\frac{W_{k}(t)}{F_{k,w,m,n}(t)\cdot \eta (t)}+\varsigma \left(t\right)\cdot \frac{\rho_{k,\;n}\left(t\right)}{M_{k}\left(t\right)},$$
where $W_{k}(t)$ represents the amount of work to be processed and $F_{k,l,m,n}\left(t\right)$ denotes the allocated computing resources. Additionally, η(t) is a coefficient representing the processing efficiency of the node, whereas ς(t) denotes for the additional computational overhead, reflecting the extra time incurred during computation. Furthermore, $\rho_{k,\;n}(t)$ represents the task wait time based on the network conditions. Finally, $M_{k}(t)$ indicates the number of cores used for parallel processing or resource availability, thereby ensuring that processing delays are managed according to the resource availability. The total service delay $\Gamma_{k}$ comprises several components and is defined as follows:(9)$$\Gamma_{k,\;total}\left(t\right)=\Gamma_{k,prep,\;trans}\left(t\right)+\Gamma_{k,\;trans}\left(t\right)+\Gamma_{k,\;queue,\;trans}\left(t\right)+\Gamma_{k,\;prep,proc}\left(t\right)+\Gamma_{k,\;queue,proc}\left(t\right)+\Gamma_{k,proc}\left(t\right).$$

These include the transmission-preparation, transmission, waiting, computation-preparation, computation-waiting, and processing delays. If the total service delay is less than the maximum allowable response time, the task is considered successfully processed. This allows for sophisticated performance evaluation that considers all possible delay factors based on the network state and available computing resources. The ratio of successfully processed tasks to the total number of processed tasks is defined as the success rate and formulated as follows(Appendix A):(10)$$\rho_{w,m,n}\left(t\right)=\frac{U_{success,w,m,n}(t)}{U_{total,w,m,n}(t)}\cdot \left(\frac{1}{1+exp(-ο\cdot (R_{w,m,n}\left(t\right)-R_{th}}\right),$$
where $U_{success,w,m,n}(t)$ denotes the numbers of successfully processed tasks and $U_{total,l,m,n}(t)$ those processed during the given time period. $R_{w,m,n}\left(t\right)$ reflects the state of the resources allocated to the *w*-th task, $R_{th}$ is the resource threshold, and ο adjusts the impact of resource status on the success rate. Finally, the success rate of each NSS is defined as follows:(11)$$\rho_{m,total}\left(t\right)=\frac{\sum_{w=1}^{W}\sum_{j=1}^{N_{w,m}}\alpha_{w,m,j}\cdot U_{success,w,m,j}(t)}{\sum_{w=1}^{W}\sum_{j=1}^{N_{w,m}}U_{total,w,m,j}(t)},$$
where $\alpha_{w,m,j}$ is a weighting coefficient that reflects the importance of each resource and can be adjusted based on the resource status or network load. By using this weighted system, more emphasis can be placed on critical resources to enhance the resource-allocation and task-processing efficiency. The weighted average of the success rate for the tasks in each NS segment is defined as follows:(12)$$\rho_{avg}\left(t\right)=\sum_{m=1}^{M}\rho_{m,total}\left(t\right)\cdot \beta_{m}(t),$$
where $\beta_{m}(t)$ is the weighting coefficient for each NSS and is set to reflect the importance and priority of each segment. It provides critical information for comparing performance across network slices and aids in establishing appropriate resource allocation and optimization strategies.

### 3.5. Problem Formulation

This study aims to address NS and resource-allocation issues in MEC networks. The objective was to maximize the weighted average completion rate of users’ computational tasks based on the total number of network and computing resources. Thus, the tasks of each network-slice entity are processed through NS and resource allocation while meeting the service requirements of each user. To achieve this, it is assumed that the network-slice entities are provided, and the focus is on the periodic resource-allocation problem for different types of computational tasks. This problem can be formulated as follows:(13)$$P:\;\underset{B,F}{\max}\underset{T\to \infty }{\lim}\frac{1}{T}=\sum_{t=1}^{T}\rho_{avg}\left(t\right),$$
and is subject to the following constraints:(13a)$$\rho_{k}\left(t\right)\leq \rho_{max},\;\forall k\in \left\{1,\;\ldots ,U_{w,m.n}\right\},\;\forall w\in \left\{1,\;\ldots ,W\right\},\;\forall m\in \left\{1,\;\ldots ,M\right\},\;\forall n\in \left\{1,\;\ldots ,N\right\},$$
(13b)$$B_{w,m}\left(t\right)\leq B_{w,m,total}\left(t\right),\;F_{w,m}\left(t\right)\leq F_{w,m,total}\left(t\right),\;\forall w,\;m,$$
(13c)$$B_{n}\left(t\right)\leq B_{n,max},\;F_{n}\left(t\right)\leq F_{n,max},\;\forall n,$$
(13d)$$\sum_{m=1}^{M}B_{w,m}\left(t\right)=B_{w,\;total}(t),\;\sum_{m=1}^{M}F_{w,m}\left(t\right)=F_{w,\;total}(t)\forall w\in \left\{1,\;\ldots ,W\right\},\;\forall m\in \left\{1,\;\ldots ,M\right\},$$
(13e)$$\sum_{n=1}^{N}B_{n}(t)\leq B_{network,\;max},\;\sum_{n=1}^{N}F_{n}(t)\leq F_{network,\;max},\;\forall n\in \left\{1,\;\ldots ,N\right\}.$$

These constraints include critical factors for ensuring the efficient allocation and management of network resources:Constraint (13a) ensures that the service delay for each user task satisfies the delay requirements of the network; thus, it is key to maintaining user experience and network performance.Constraint (13b) ensures that each network slice operates efficiently with the resources allocated to it, thereby preventing resource wastage and enhancing system efficiency.Constraint (13c) guarantees that resources are not overused by each network node, thereby maintaining network stability and reliability.Constraint (13d) balances resource allocation across network-slice entities, prevents the concentration of resources at specific slices, and ensures fair resource distribution.Finally, Constraint (13e) ensures that total network-resource usage does not exceed available resources and manages and optimizes the overall network performance.

These constraints effectively address the NS and resource-allocation issues in an MEC network and play a key role in maximizing its performance and stability.

## 4. RL-Based NS and Resource-Allocation Algorithm

### 4.1. Overview

Because ECS and ARS operate on different time scales, solving the entire problem using existing algorithms is challenging. Therefore, it is decomposed into two subproblems: ECS (SP1) and ARS (SP2). SP1 periodically conducts ECS for each network-slice entity to maximize the weighted average completion rate of users’ computational tasks. This ensures the efficient use of network resources and optimizes the user experience and can be expressed as follows:(14)$$SP1:\;\underset{\mathrm{\pi ECS}}{\max}E\left[\sum_{t=1}^{T}\alpha_{new}\cdot \upsilon ECS(t)\right],$$
where πECS represents the resource-allocation policy for conducting ECS within a network-slice instance and υECS(t) denotes the reward calculated at time *t* based on the resource-allocation efficiency and success rate of user computational tasks. This equation reflects the essence of ECS and provides a direction for maximizing NS effectiveness.

By contrast, the objective of SP2 is to allocate resources for each computational task such that network-resource utilization is maximized at the ENs associated with the service-level slice instances. This problem can also be addressed using RL and is formulated as follows:(15)$$SP2:\underset{\mathrm{\pi ARS}}{\max}E\left[\sum_{t=1}^{T}\upsilon ARS(t)\right],$$
where πARS represents the resource-allocation policy for conducting ARS at the ENs and υARS(t) denotes the reward calculated at time *t* based on the network-resource utilization. This equation aims to optimize resource utilization at the ENs, enhance overall network performance, and minimize resource wastage.

Therefore, we propose two algorithms to solve the network slicing and computing resource-allocation problems: MADDPG-based ECS and SAC-based ARS. The ECS algorithm optimizes resource scheduling for each network-slice entity, whereas the ARS algorithm implements adaptive resource allocation for each service-slice instance. The integration of these algorithms constitutes the overall MAARS framework, which leverages a multiagent system based on the actor–critic structure to efficiently manage network resources, thereby optimizing overall performance. It aims to maximize resource utilization while minimizing resource conflicts between network slices and services, enabling tailored resource allocation and scheduling based on the characteristics of each instance. Thus, it offers a robust solution that not only enhances network performance but improves overall system quality (QoS) through efficient resource utilization.

### 4.2. MADDPG Algorithm

MADDPG is a multiagent RL algorithm that aims to learn the policy ($\pi_{i}$) and *Q*-function ($Q_{i}$) of each agent. The algorithm employs centralized training and decentralized execution, wherein a central controller trains an agent-network group based on global environmental information, whereas the agents select and execute actions based on local environmental information [29,30]. Decentralized execution mitigates the problem of the joint-action space growing exponentially as the number of agents increases; therefore, traditional single-agent RL algorithms are unsuitable for this task.

At each step of the MADDPG algorithm, agents first obtain local observations $o_{j}(t)$ of the environment, select actions $a_{j}(t)$ based on $o_{j}(t)$, where j represents the agent index, and execute the selected actions to receive a reward *r(t)*. After executing the selected actions, the environment is updated, and the agents receive new local observations $o_{j}(t+1)$ in the next step. The central controller collects the new observations of all agents and stores the experience (*s*(*t*), *a*(*t*), *r*(*t*), *s*(*t +* 1)) in a replay buffer, where *s*(*t*) and *s*(*t +* 1) denote the vectors containing the observations of all agents at time steps *t* and *t* + 1, respectively. A batch of experiences is then sampled from the replay buffer to train the networks for all agents in the MADDPG algorithm [31].

MADDPG uses a set of agent networks and a central critic network to learn policy and *Q*-function $Q_{i(o,a_{i})}$ of each agent as follows:

Agent networks: These networks employ a DDPG structure, which includes both the evaluation and target networks, and are used to learn the policies $\pi_{i}$ and *Q*-functions of each agent. The input for the agent networks comprises the current observation $o_{j}(t)$, and they output the action $a_{j}\left(t\right)$ or *Q*-value $Q_{j(o,a_{j})}$.

Central critic network: This network takes the observations and actions of all agents as inputs and outputs the *Q*-value for the *Q*-function of each agent. Using this information obtained from all agents, the central critic network computes the *Q*-values that allow each agent to learn a better policy.

Similar to the DQN, MADDPG trains the parameters of all agent networks by minimizing a loss function, which is defined as follows:(16)$$L\left(\theta_{i}\right)=E_{\left(o,a,r,o^{\prime }\right)}\left[{\left(y_{i}-Q_{i}(o,a_{1},\ldots ,a_{N};\theta_{i}\right)}^{2}\right],$$
(17)$$y_{i}=r_{i}+\gamma Q_{i}^{\prime }\left(o\prime ,a_{1}^{\prime },\ldots ,a_{N}^{\prime };\theta_{i}^{\prime }\right),$$
where $Q_{i}^{\prime }$ and $\theta_{i}^{\prime }$ denote the target network and its parameters, respectively, $\theta_{i}$ denotes the parameters of the evaluation network, and $a_{N}^{\prime }$ is the action selected by the target network.

The MEC-network environment can be reflected by the current number of user computational tasks, network state, and computing resources available for each network-slice entity. The virtual controller of each network-slice entity can only observe the local environment and manage its network-slice entity. Thus, the network model can be considered a decentralized execution model. However, each network-slice entity is interdependent. Because the total number of network and computing resources is limited across all ENs and all M logical access networks share the same resources, ECS must be conducted using a centralized training mode to obtain an optimal slicing policy.

Thus, the MADDPG algorithm, which employs centralized learning and decentralized execution, is well-suited for solving SP1 to achieve an optimal policy. Compared to the independent Q-learning (IQL) algorithm, MADDPG can enhance the relationship between the policy $\pi_{i}$ and *Q*-function $Q_{i(o,a_{i})}$ of each agent [32]. The IQL algorithm involves independent learning by each agent, which can lead to poor performance in environments where interaction is important. By contrast, the MADDPG algorithm considers agent interactions through a central critic, thereby enhancing performance.

#### MADDPG-Based ECS Algorithm

The MADDPG-based ECS algorithm efficiently manages core-to-edge resource allocation in NS environments, optimizing resource utilization to maximize task success rates. By leveraging MADDPG, the ECS layer effectively addresses the challenges of coordinating resource distribution across the network hierarchy, enabling a dynamic and adaptive response to changing demands.

To accelerate learning and further enhance performance, we employ MAML for initialization. This significantly reduces the convergence time of MADDPG, allowing the ECS algorithm to adapt quickly to evolving network conditions and ensuring consistent performance. Additionally, its integration with the SAC-based ARS layer ensures that the outputs from ECS seamlessly guide real-time resource slicing, creating a cohesive and efficient optimization loop that spans the entire network. This two-tiered approach ensures coordinated resource management from the core to the edge.

Compared to traditional single-agent or static optimization methods, our MADDPG-based ECS algorithm offers significant advantages. It provides improved resource efficiency, dynamic adaptability to complex and evolving demands, and higher task-completion rates. Moreover, it offers a robust and scalable solution for optimizing resource allocation for MEC NS. The agents, states, actions, and reward functions are defined as follows:Agent: The agent acts as a virtual controller for each network-slice entity periodically performing ECS. Each agent monitors the resource status of its respective slice instance and makes optimal resource-allocation decisions.State: The state is represented by a set of local observations of all agents. Each observation includes the resource and task status obtained from the network-slice entity and is defined as follows:(18)$$o_{m}\left(t\right)=\left\{{SE}_{w,m,n}\left(t\right),{RN}_{w,m,n}\left(t\right),\;{SR}_{w,m,n}(t),\;{PU}_{w,m,n}(t)|n=1,\;2,\;\ldots ,\;N_{m},w=1,\;2,\ldots ,\;W\right\},$$
where ${SE}_{w,m,n}\left(t\right)$ represents the number of available NRU resources at the *n*-th BS of the slice entity and ${RN}_{w,m,n}\left(t\right)$ indicates the number of available CRU resources at the n-th EN. Additionally, ${SR}_{w,m,n}(t)$ denotes the task-completion rate at the *n*-th edge node, ${PU}_{l,m,n}\left(t\right)$ represents the number of user computational tasks per time unit, and $N_{m}$ indicates the number of BS/EN. Based on the local observations of each agent, the state is defined as follows:(19)$$s\left(t\right)=\left\{o_{m}\left(t\right)|m=1,\;2,\;\ldots ,\;M\right\}.$$Action: Each agent performs ECS within its respective network-slice entity. Specifically, it can choose to increase, decrease, or maintain the number of NRUs or CRUs in each BS. The action space of each agent is expressed as follows:
(20)$$a_{m}\left(t\right)=\left\{{SV}_{w,m,n}\left(t\right),SW_{w,m,n}\left(t\right)|n=1,\ldots ,N_{m},w=1,\ldots ,W\right\},$$
where ${SV}_{w,m,n}\left(t\right)=1$ and ${SW}_{w,m,n}\left(t\right)=1$ represent increases in the NRU and CRU resources, respectively, at the n-th edge server of the network-slice entity. Conversely, $SV_{w,m,n}\left(t\right)=-1$ and ${SW}_{w,m,n}\left(t\right)=-1$ indicate decreases in the NRU and CRU resources, respectively, whereas $SV_{l,m,n}\left(t\right)=0$ and $SW_{l,m,n}\left(t\right)=0$ signify no changes in resources. The number of resource blocks that can increase or decrease at each step is fixed. The joint-action vector of the MADDPG algorithm is defined as follows:(21)$$a\left(t\right)=\left\{a_{m}\left(t\right)|m=1,\;2,\;\ldots ,\;M\right\}.$$Reward Function: Because the goal of the algorithm is to maximize the long-term weighted average of the task-completion rate, a penalty-based reward function was designed. If the task-completion processing rate falls below a certain threshold, a penalty is applied to reduce the reward. This reward structure encourages higher performance by penalizing agents that lead to poor performance, thereby driving them to achieve better results. The reward function is defined as(22)$$r_{m}\left(t\right)=\left\{\begin{array}{c} \;\;\omega_{2}\cdot \chi_{m}\left(t\right)+{\bar{\varpi }}_{2}\;\;if\;\chi_{m}\left(t\right)\geq \;\chi_{threshold}\;\\ \;\;\omega_{2}\cdot \chi_{m}\left(t\right)+{\bar{\varpi }}_{2}-\Gamma \;if\;\chi_{m}\left(t\right)<\chi_{threshold} \end{array}\right.$$
where, $\omega_{2}>0$ is a coefficient that determines the reward slope based on the task-completion rate and ${\bar{\varpi }}_{2}$ sets the offset value of the reward. Additionally, $\chi_{threshold}$ is the threshold of the task-completion rate and Γ>0 is the penalty applied when the performance falls below the threshold.


Figure 2 illustrates the structure of the MADDPG-based MAARS algorithm, where multiple agents interact with network-slice entities to optimize resource allocation. Each agent utilizes actor and critic networks to collaboratively manage resources and ensure efficient task processing across network slices. Algorithm 1 summarizes the main steps of the MADDPG-based ECS algorithm, which represents an approach wherein each agent acts independently but cooperates with other agents through centralized learning to efficiently allocate resources in an NS environment.
**Algorithm 1: MADDPG-based ECS**1: Initialize policy networks $\pi_{i}$ and *Q*-functions $Q_{i}$ for all agents2: Initialize target networks ${\pi \prime }_{i}$ and ${Q\prime }_{i}$ with the same parameters3: Initialize replay buffer R4: **for** episode = 1 to ${NP}_{epis}$:5: Initialize random exploration process6: Initialize state $s_{0}$7: **for** step = 1 to ${NQ}_{step}:$**8**: for each agent i:9: Observe local state $o_{i}(t)$ and select action $a_{i}\left(t\right)=\pi_{i}\left(o_{i}\left(t\right)\right)+noise$10: Execute joint-action *a(t)* = {$a_{1}\left(t\right),\;\ldots ,\;a_{M}(t)$}11: Observe reward $r_{m}(t)$ and next state *s(t +* 1*)*12: Store experience *(s(t)*, *a(t)*, $r_{m}(t)$, *s(t +* 1*)* in R13: Sample mini-batch from R, for each agent i:14: Compute target action $a_{i}(t+1)$ and target *Q*-value $y_{i}$15: Update $Q_{i}$ by minimizing L($\theta_{i})$16: Update $\pi_{i}$ using policy gradient17: Soft update target networks ${\theta \prime }_{i}$ and ${\theta \prime }_{Q_{i}}$18: Decay exploration noise for next episode

The policy network and *Q*-function for each agent are initialized, and the same parameters are set for the target network. A random exploration process is initialized at the start of each episode. In each step, the agent observes the local state and selects an action. After executing the selected action, the agent receives a reward, observes the next state and the reward, and stores the experience in the replay buffer. Next, the *Q*-function is updated using a mini-batch sampled from the replay buffer, and the policy network is updated based on the policy gradient. Finally, the target network is soft-updated, and the exploration noise is reduced for the next episode. This algorithm aims to maximize the efficiency of network-resource allocation and optimize the task-completion rate.

### 4.3. MAML-Based Initialization Algorithm 

For a given weight vector, the MADDPG-based ECS algorithm periodically performs ECS to learn the optimal policy aligned with that vector. However, when the weight vector changes, relearning the optimal policy can be time-consuming [33,34,35]. To address this, the MAML-based initialization algorithm is introduced, which obtains a meta-policy a set of optimal initialization parameters for the agent, mixture, and hyper-networks employed in the MADDPG-based algorithm. This enables faster adaptation to new weight vectors, thereby significantly reducing the time required to re-learn the optimal policy.

#### 4.3.1. Meta-Policy of MAML-Based Initialization Algorithm

The meta-policy provides an initial optimized state that can quickly adapt to various objectives with different weight vectors. This serves as an optimal initial policy that can be effectively trained for diverse objectives [36]. The MAML-based initialization algorithm is model-agnostic and can learn a meta-policy that swiftly adapts to different goals.

By providing a robust initialization, MAML accelerates the convergence of the MADDPG-based ECS algorithm, ensuring better performance, even under dynamic network conditions. The meta-policy allows agents to start from a well-optimized state, reducing the iterations needed to achieve stable policies. This is crucial in MEC networks, where rapid adaptation is essential for handling fluctuating resource demands and maintaining service quality.

In this algorithm, the meta-policy parameters are updated based on the optimal policy parameters for each objective. This algorithm comprises inner and outer loops. In the inner loop, the policy parameters are trained for a specific objective to determine the optimal policy, and gradient descent is employed to minimize the loss function of the MADDPG-based ECS algorithm. In the outer loop, the gradients of each optimal policy parameter are calculated to update the meta-policy parameters.

Integrating MAML into MADDPG enhances the algorithm’s ability to handle diverse and dynamic network environments with improved efficiency and adaptability. This capability is crucial for optimizing resource allocation and enhancing the overall performance of MEC networks.

#### 4.3.2. MAML-Based Initialization Algorithm for Policy Optimization

The MAML-based initialization algorithm comprises the following three main stages:Adaptation: A set of weight vectors $A=\left[a_{1},\;a_{2},\ldots ,\;a_{i}\;\right]$ is given, where *i* represents the total number of elements and each vector contains a series of weighting coefficients. SP1 determines the optimal policy for each weight vector. To achieve this, the policy is periodically updated using each weight vector. In the inner loop, the policy parameters $\phi_{i}$ are optimized for each weight vector $a_{i}$ through gradient descent as follows:(23)$$\phi_{i}^{\prime }=\phi_{i}-\omega_{3}\nabla_{\phi_{i}}L_{a_{i}}(\phi_{i}),$$
where $\phi_{i}^{\prime }$ and ${\bar{\varpi }}_{3}$ denote the updated parameters and learning rate, respectively. Through this process, the optimal policy parameters $\phi_{i}^{*}$ for each weight vector are obtained as follows:(24)$$\phi_{i}^{*}=arg\;\underset{\phi_{i}}{\min}L_{a_{i}}(\phi_{i}).$$Meta-policy Training: In this stage, the optimal policy parameters $\phi_{i}^{*}$ obtained in the previous stage are used together to learn the optimal meta-policy parameters. In the outer loop, the optimal policy parameters for various weight vectors from the inner loop are used to update the meta-policy parameters as follows:
(25)$$\Theta =\Theta -\iota \sum_{i}\nabla_{\Theta }L_{a_{i}}\left(\phi_{i}^{*}\right),$$
where ι represents the metalearning rate. Through this process, Θ of the meta-policy are optimized.Fine-tuning: To achieve the objective of SP1 with a new weight vector $\alpha_{new}$, the meta-policy parameters are used as the initial parameters for the agent, mixture, and hyper-networks in the MADDPG-based algorithm. Subsequently, by iteratively fine-tuning the initial policy using this algorithm, the optimal resource-allocation policy $\pi_{ECS}$ aligned with $\alpha_{new}$ can be obtained. This process aims to maximize ECS performance by efficiently allocating resources within the network-slice instance while maintaining or enhancing the completion rate of user computational tasks. This stage includes updating the policy parameters by minimizing the loss function as follows:(26)$$\Theta_{new}^{\prime }=\Theta -\mu \sum_{i}\Theta L_{a_{i}}\left(\phi \right),$$
where μ is the learning rate during the fine-tuning stage. The MAML-based initialization algorithm offers several benefits. First, it allows for rapid learning of an optimal policy for new weight vectors through the meta-policy, thereby enabling quick adaptation to various environmental changes. Second, it minimizes overall training time by reducing the time required to re-learn the policy using initialized parameters. This is particularly beneficial in scenarios that require repetitive learning. Third, it can create a policy that works effectively across different weight vectors, thereby increasing system generalizability. These advantages help improve overall system performance and maximize efficiency. Algorithm 2 summarizes the key steps of the MADDPG-based ECS algorithm, which aims to learn a set of initialized parameters that can be used to quickly determine the optimal policy for various weight vectors.
**Algorithm 2: MAML-based Initialization**1: Initialize meta-policy parameters Θ2: **for** iteration = 1 to ${num}_{iterations}$:3: Sample a batch of weight vectors $\left\{a_{1},\;a_{2},\;\ldots ,\;a_{B}\right\}$ from A4: **for** each weight vector $a_{i}$ perform:5: Initialize policy parameters $\varphi_{i}$ with Θ6: Update $\varphi_{i}$ using gradient descent on $L_{a_{i}}(\phi_{i}^{*})$ over K steps7: Store the optimal policy parameters $\varphi_{i^{*}}$8: **end for**9: Compute meta-policy update: $\Delta \Theta =-\eta \sum_{i}\nabla_{\Theta }L_{a_{i}}(\phi_{i}^{*})$10: Update meta-policy parameters: Θ=Θ+ΔΘ11: **end for**12: Fine-tune policy parameters Θ using new weight vector $\alpha_{new}$


In the first step of the algorithm, the initial meta-policy parameters are set. These parameters are trained to provide optimal initialization across multiple weight vectors. Subsequently, the meta learning loop begins, wherein the meta-policy is learned over various iterations. In each iteration, various weight vectors are sampled, and the optimal policy parameters are learned based on them. For each batch of sampled weight vectors, the policy parameters are initialized based on a meta-policy. In the inner loop, these parameters are optimized through gradient descent. The optimized policy parameters are then stored and used to update the meta-policy.

Finally, when a new weight vector is provided, the algorithm quickly fine-tunes the optimal policy based on the parameters initialized by the meta-policy. This process continuously enhances the meta-policy, enabling rapid adaptation to new environments.

### 4.4. SAC-Based Resource-Allocation Management (RAM) Algorithm

The SAC-based RAM algorithm plays a critical role in efficiently managing network resources at the edge, providing the flexibility and scalability required to meet diverse user demands. A policy-based optimization algorithm known for its high sample efficiency and stability [37], SAC is particularly well-suited for the dynamic resource management challenges inherent in MEC networks. Within the MAARS framework, SAC complements the MADDPG-based ECS algorithm by focusing on real-time, fine-grained resource slicing at the network edge, guided by the outputs of ECS.

This dual-layer approach combines the global coordination capabilities of MADDPG with the precise local optimization of SAC. MADDPG handles the core-to-edge resource allocation, while SAC refines this allocation at the edge, providing a comprehensive solution to the challenges of resource management in MEC networks. This integration ensures seamless and efficient resource allocation across the entire network hierarchy. Compared to traditional methods, the SAC-based RAM algorithm offers significant advantages. It not only improves resource utilization and task success rates but dynamically adapts to the ever-changing demands of the network. The combination of MADDPG and SAC delivers a robust and scalable solution for MEC network slicing, ensuring superior performance and adaptability even in the most complex and dynamic environments.

The learning process of this algorithm comprises multiple iterations. During training, experience replay and target networks are used to increase stability. Experience replay improves sample efficiency and reduces data correlation by using a buffer to store the experiences of each agent [38]. The target network is periodically updated, which contributes to the stability of the learning process.
Agent: The agent is responsible for making resource-allocation decisions and corresponds to network nodes such as the BS or edge servers. Each agent allocates resources based on the network state and learns resource-management policies while providing services to users.State: The state comprises a set of parameters that represent the current network conditions. This state information is essential for an agent to understand the environment and optimize resources. The state variables are defined as follows:
(27)$$State\left(t\right)=\left\{{Resource}_{Usage}\left(t\right),\;{Queue}_{Length}\left(t\right),\;{Demand}_{User},{Delay}_{Max}\;\right\},$$
where ${Resource}_{Usage}\left(t\right)$ represents the number of resources currently in use and indicates the ratio of allocated resources to the total resources available in the network. ${Queue}_{Length}\left(t\right)$ reflects the queue length, which indicates the number of tasks waiting to be processed in the network. Additionally, ${Demand}_{User}$ refers to the number of resources that must be allocated to each user in the network and ${Delay}_{Max}$ represents the maximum allowable delay to deliver each service to the users. These state variables play a critical role in network-resource management by providing essential information for the agent to make optimal resource-allocation decisions.Action: The action of an agent involves determining the number of resources that must be allocated for each user request based on the current state and is defined as follows:
(28)$$Action\left(t\right)=\{a_{i}(t)|i\in \{1,2,\ldots ,{Count}_{User}(t)\}\},$$
where $a_{i}(t)$ represents the number of resources allocated to user *i* at time *t*. $\in \{1,2,\ldots ,{Count}_{User}(t)\}$ represents the total number of users connected to the network. The agent makes resource-allocation decisions for each user. Therefore, $Action\left(t\right)$ is an action vector that represents the resource allocation for all users, thereby playing a critical role in efficient network-resource management and meeting user demands.Reward: The reward function evaluates the resource-allocation efficiency and guides the agent to learn better policies. It is defined as the difference between the actual and target resource usage, as follows:
(29)$$reward\left(t\right)=\sum_{i=1}^{{Count}_{User}(t)}\left(\frac{\vartheta_{max}-\vartheta_{actual,\;i}(t)}{\vartheta_{max}}\cdot I\left(\vartheta_{actual,\;i}\left(t\right)\leq \vartheta_{max}\right)-I\left(\vartheta_{actual,\;i}\left(t\right)>\vartheta_{max}\right)\right).$$


This function evaluates the resource-allocation efficiency at a specific time *t*. If the actual transmission delay $\vartheta_{actual,i}$ for each user *i*, is within the maximum allowable delay $\vartheta_{max}$, the reward is maintained as $\frac{\vartheta_{max}-\vartheta_{actual,i}(t)}{\vartheta_{max}}$. This encourages efficient resource allocation by granting higher rewards for shorter delays. Conversely, if $\vartheta_{actual,i}\left(t\right)\leq \vartheta_{max}$, a penalty is applied to that agent. The total reward is calculated by summing these values across all users, thereby indicating the overall efficiency of network-resource allocation. This reward structure aims to optimize resource allocation and guide the agent to minimize delays during learning.

#### Learning Process of SAC-Based RAM Algorithm

The SAC-based RAM algorithm enhances learning stability by using experience replay and target networks. Experience replay includes a buffer that stores user experiences, enhances sample efficiency, and minimizes data correlation. The target network is only updated periodically to further enhance the stability of the learning process. The pseudocode of the learning process of this algorithm is presented in Algorithm 3.
**Algorithm 3: SAC-based RAM**1: Initialize actor-network parameters $\theta_{p}$ and critic-network parameters ϕ2: Initialize target-network parameters φ′ with φ3: **for** episode = 1 to ${NP}_{epis}$ perform:4: for t = 1 to ${NP}_{epis}$ perform:5: Observe current state $s_{t}$6: Select action $a_{t}$ using policy π($\theta_{p}$|$s_{t}$)7: Execute action $a_{t}$ to receive reward $r_{t}$ and next state $s_{t+1}$8: Store ($s_{t}$, $a_{t}$, $r_{t}$, $s_{t+1}$) in replay buffer9: Sample a random mini-batch from replay buffer10: Compute temporal-difference error $\delta_{t}$:11: $\delta_{t}$ = $r_{t}$ + γ × $Q_{\varphi \prime }$ ($s_{t+1}$, π($\theta_{p}$|$s_{t+1}$)) − $Q_{\varphi }$($s_{t}$, $a_{t}$)12: Update critic network by minimizing:13: L(φ) = (1/2) × ∑($\delta_{t^{2}}$)14: Update actor network using:15: $J_{SAC}$($\theta_{p}$) = $E_{t}$ [min($\psi_{t}$($\theta_{p}$), clip($\psi_{t}$ ($\theta_{p}$), 1 − ϵ, 1 + ϵ)) ×$\delta_{t}$]16: Update target network ϕ’:17: **if** t % target_update_interval == 0:18: φ′ ⟵ φ

This algorithm focuses on maximizing the efficiency of network-resource allocation and learning an optimal resource-management policy. It enables stable and flexible resource allocation under various network conditions and can adapt to network performance fluctuations. The SAC-based approach is particularly effective in high-dimensional continuous action spaces and offers robust performance for solving complex resource-allocation problems. Moreover, the SAC algorithm provides high sample efficiency, allowing the agent to quickly learn optimal resource-allocation policies, even with limited training data. Consequently, it reduces the complexity of network-resource management while ensuring optimal resource utilization and improving the QoS.

### 4.5. Computational Complexity

The computational complexity of the proposed MAARS framework is determined by the operations in Algorithms 1, 2, and 3, each contributing to the overall efficiency and scalability of the system. Algorithm 1, the MADDPG-based ECS algorithm, primarily involves nested loops over tasks and resources. This results in a time complexity of O(N×M), where N represents the tasks and M represents the resources, with a space complexity of O(N) for task states. Algorithm 2, the MAML-based initialization algorithm, incorporates matrix operations for optimizing the meta-policy parameters. This leads to a time complexity of $O(N\times M+M^{2})$ and a space complexity of $O(N+M^{2})$ for the resource matrices. Algorithm 3, the SAC-based RAM algorithm, handles RL-based policy updates for resource allocation. The time complexity of this algorithm is O(N×M), where N represents the number of slices and M represents the number of iterations in the learning process, and a space complexity of O(N+P), where P represents the policy parameters. By combining the complexities of these individual algorithms, the overall MAARS framework achieves a time complexity of $O(N\times M+M^{2})$ and a space complexity of $\left(N+M^{2}+P\right)$. This analysis demonstrates that the proposed framework is designed for scalability and efficiency, making it suitable for deployment in large-scale MEC networks with numerous tasks, resources, and network slices.

### 4.6. Performance Evaluation

The performances of the proposed algorithms were validated through simulations. First, to evaluate the SAC-based RAM algorithm, we compared its performance with that of the random resource-allocation [39], proportional fairness (PF) [40], and centralized resource-allocation (CRA) [41] algorithms. The random resource-allocation algorithm randomly allocates resources to user tasks, whereas the PF algorithm allocates them fairly based on both resource availability and user demands. By contrast, the CRA algorithm manages all network resources centrally and aims to achieve optimal resource distribution.

Next, the proposed MAARS algorithm was evaluated by comparing its efficiency and performance with those of heuristic-based, DQN-based, and advantage actor–critic (A2C)-based resource-allocation algorithms. The heuristic-based algorithm allocates resources based on predefined rules, the DQN-based algorithm allocates them based on the data size and computation requirements of a task, and the A2C-based algorithm conducts periodic resource allocation using the A2C algorithm.

To further assess the performance of the MAARS framework, we conducted a comprehensive analysis of the impact of MAML. By leveraging MAML to initialize the ECS neural networks with pretrained meta-policy parameters, we observed a significant reduction in the number of training iterations required for convergence. This advanced initialization strategy enabled the ECS component to adapt more effectively to dynamic network conditions and fluctuating resource demands, ultimately enhancing the overall performance and scalability of the framework.

The evaluation metrics included the success rate for user tasks within the service-slice entities and the weighted average of the success rates across all network-slice entities. To comprehensively evaluate the contribution of MAML, we analyzed additional metrics, such as convergence time and resource-allocation adaptability. The results demonstrated faster convergence and more stable resource utilization, particularly under dynamic network conditions, highlighting the effectiveness of MAML in enhancing the adaptability and efficiency of the framework. The simulations were implemented using Python (version 3.12.4) and PyTorch (version 2.4), assuming that user-task arrivals followed a Poisson process. The system configuration used for the simulations included an Intel Core i7-9700K CPU (Santa Clara, CA, USA) running at 3.60 GHz, 32 GB of DDR4 RAM, and an NVIDIA RTX 2080 Ti GPU (Santa Clara, CA, USA). The key simulation parameters are listed in Table 2.

## 5. Results

### 5.1. SAC-Based RAM Algorithm

Figure 3 compares the task-completion ratios at the BS within the service-slice entity and the task-arrival rates for the SAC-based RAM and the adaptive and random resource-allocation algorithms. As the task-arrival rate increased, the completion rates of all algorithms gradually decreased. This is because as more computational tasks arrive at the BS under a higher arrival rate, processing delays occur, leading to a higher percentage of tasks failing to meet the maximum allowable service delay. However, the proposed SAC-based RAM algorithm demonstrates a better task-processing rate than the other two algorithms.

The SAC-based RAM algorithm exhibits a higher task-completion ratio because it dynamically allocates both network and computing resources by simultaneously considering the task demand and current resource state of the BS. This allows it to efficiently allocate resources, even under heavy loads, thereby minimizing task-processing delays. Specifically, by continuously monitoring network conditions in real-time and optimally readjusting resource availability, unnecessary wait times are minimized, and the task-completion rate is increased. Thus, the SAC-based RAM algorithm provides stable performance even if the network load increases, thereby maximizing resource efficiency.

Figure 4 compares the task-completion ratios of the SAC-based RAM, adaptive, and random algorithms at the BS based on the CPU frequency. As the CPU frequency increases, the task-completion ratios of all algorithms initially increase before stabilizing. This trend occurs because a higher CPU frequency increases the computing resources that can be allocated to tasks, thereby enhancing the likelihood of processing tasks within the maximum allowable service delay. Although the task-completion ratios of all algorithms increased, that of the SAC-based RAM was the highest.

This is because the design of the proposed RAM algorithm is based on the SAC framework, which enables dynamic resource allocation and CPU utilization optimization, ensuring that appropriate computing resources are allocated to each task in real-time. The SAC-based RAM algorithm maintains resource efficiency by considering both CPU status and user-task demands, even when the CPU frequency increases. This highlights that merely increasing the CPU frequency has limitations in improving the task-completion ratio, underscoring the importance of resource management and allocation optimization.

Figure 5 compares the task-completion ratios of the SAC-based RAM, adaptive, and random algorithms at the BS based on bandwidth. As the bandwidth increased, the task-completion ratios of all algorithms initially increased before stabilizing. This pattern occurs because a larger bandwidth allows more spectrum resources to be allocated to the tasks, increasing the likelihood of processing them within the maximum allowable delay. However, once sufficient bandwidth is allocated, the processing capacity becomes saturated, and additional bandwidth increases have a lower impact on performance.

Nevertheless, the SAC-based RAM algorithm maintains a higher task-completion ratio than the other algorithms through real-time resource optimization and efficient allocation. This is because the RL-based approach dynamically analyzes network conditions and task demands to allocate resources appropriately.

Therefore, the proposed SAC-based RAM algorithm demonstrates superior performance to the other algorithms under key resource conditions, such as arrival rate, CPU frequency, and bandwidth, and maintains a higher task-completion ratio even under resource-overload conditions. Owing to its dynamic optimization and real-time adaptability, the SAC-based RAM algorithm can play a crucial role in more complex network environments in the future. Consequently, it can be essential for maximizing network-resource utilization and maintaining QoS.

### 5.2. MADDPG-Based MAARS Algorithm

Figure 6 illustrates the relationship between the reward values and the number of iterations for the MADDPG-based MAARS algorithm under conditions of both ample and scarce network and computing resources. As the number of iterations increases, the reward values converge in both environments, indicating that the algorithm progressively learns to allocate resources more efficiently and reaches a stable performance level. In environments with ample resources, the MAARS algorithm achieves higher reward values, reflecting its effectiveness in optimizing resource allocation through efficient utilization. The MADDPG-based distributed resource-management technique allows the MAARS algorithm to cooperatively allocate resources across network slices, ensuring that the needs of each slice are met optimally.

Moreover, even in resource-scarce situations, MAARS maintains its performance efficiency by adapting to network-state changes and implementing real-time resource allocation. This helps prevent excessive usage or shortages of resources. Thus, the MAARS algorithm demonstrated stable and high performance across various environments.

Figure 7 illustrates the relationship between the task-completion ratios and task-arrival rate for the MAARS, DQN–PPO, PF, CRA, and heuristic-based resource-allocation algorithms. Evidently, as the arrival rate increases, the task-completion ratios of all algorithms decrease owing to increased wait times and service delays. However, both the MAARS and DQN–PPO algorithms maintain higher task-completion ratios than the PF, CRA, and heuristic algorithms. Specifically, MAARS leverages the MADDPG-based collaborative resource-allocation technique to dynamically adjust the resources for each slice, thereby maximizing performance.

Figure 8 illustrates the task-completion ratio as a function of the CPU frequency. Evidently, as the CPU frequency increases, the completion ratio of all algorithms increases because a higher CPU frequency enhances the task-processing speed. However, the MAARS algorithm exhibits the most significant performance improvement owing to its collaborative resource-allocation feature, which efficiently distributes resources based on the requirements of each network slice. Additionally, although the performance of DQN–PPO also increases with the CPU frequency, it does not reach the level of MAARS. By contrast, the PF, CRA, and heuristic algorithms show relatively lower performance owing to limitations in their resource-allocation optimization strategies, whereas MAARS maximizes efficiency in this scenario.

Figure 9 illustrates the task-completion ratio as a function of the bandwidth. As the bandwidth increases, the task-completion ratio of all algorithms improves before stabilizing after a certain bandwidth threshold. MAARS and DQN–PPO exhibit higher performances than the PF, CRA, and heuristic algorithms because MAARS offers more efficient and collaborative resource management and allocation. At the same time, DQN–PPO benefits from combining the strengths of both deep Q-networks and proximal policy optimization, leading to more effective decision-making in dynamic environments.

MAARS is designed to reflect the real-time requirements of network slices, ensure timely resource allocation, and maximize resource utilization without wastage. By contrast, DQN–PPO manages resources independently for each slice, which limits its optimization compared with MAARS. Moreover, PF focuses on fairness, which restricts performance optimization, whereas CRA struggles to adapt quickly to real-time changes owing to its centralized approach. Additionally, the heuristic-based algorithm uses fixed resource-allocation rules, failing to adapt to complex network demands. Therefore, MAARS efficiently utilizes resources even as the bandwidth increases, maintaining a higher task-completion ratio than the other algorithms.

Figure 10 shows the utility-function values for three network-slice instances obtained using the MADDPG-based MAARS algorithm with respect to various weight vectors. Each weight vector represents a specific objective function, with the “initial” value indicating the utility before NS. The MAARS algorithm adopts a collaborative distributed resource-management approach for each network slice, leading to more efficient resource allocation. Assigning higher weights to specific network slices increases their utility-function values, reflecting the capability of MAARS to intelligently adjust resource-allocation priorities among network slices. By reflecting the real-time demands of each slice, MAARS can allocate resources in a timely manner and maximize performance. Compared to traditional algorithms, such as DQN–PPO, MAARS demonstrates superior performance owing to its collaborative resource allocation, which minimizes resource wastage and ensures that optimal performance is maintained.

Figure 11 illustrates the loss ratios as the number of iterations increases for various resource-allocation algorithms, including MAARS, DQN–PPO, PF, CRA, and heuristic-based. As the number of iterations increases, the loss ratio of MAARS decreases; thus, it outperforms all other algorithms in terms of convergence speed and efficiency. DQN–PPO also demonstrates significant loss reduction, but lags behind MAARS. PF and CRA show more moderate loss reduction, whereas the heuristic algorithm exhibits the highest loss ratio. MAARS achieves superior performance owing to its collaborative and adaptive resource management, which ensures efficient task processing and resource allocation across network slices. By contrast, DQN–PPO, although effective, allocates resources independently for each slice, which limits its overall optimization performance. Furthermore, PF focuses on fairness, which constrains its ability to optimize performance, while CRA struggles to adapt quickly to dynamic network conditions owing to its centralized approach. Finally, the heuristic-based algorithm relies on fixed rules and, therefore, cannot effectively adapt to the complexity of real-time network demands. Consequently, MAARS consistently demonstrated superior loss-reduction performance over time than the other algorithms.

## 6. Discussion

This study introduced the MAARS algorithm to effectively address the challenges of resource allocation and NS in MEC networks. MAARS leverages collaborative resource allocation to achieve superior performance than existing algorithms, dynamically adapting to network changes in real-time.

The simulation results validated the efficiency of the MADDPG-based resource management technique, which efficiently allocates resources based on the real-time demands of each network slice. The integration of MADDPG and SAC creates a powerful dual-layer optimization framework. MADDPG manages global resource coordination, while SAC provides real-time adaptability at the network edge. This approach enhances scalability and operational efficiency across a wide range of network conditions. Future research should explore the potential of the MAARS algorithm in even more complex and heterogeneous network environments, further expanding its applicability for resource allocation in challenging scenarios.

## 7. Conclusions

This study presented a novel two-stage resource slicing and allocation model to address the critical challenges of NS and resource allocation in MEC networks. This model, formulated as a long-term optimization problem, aims to maximize the completion rate of user-computing tasks while effectively meeting the diverse service requirements of individual network slices. To achieve this, we decomposed resource management into two distinct yet interconnected algorithms: ECS and ARS, integrated within the proposed MAARS algorithm. To enhance the efficiency of the MADDPG-based ECS algorithm, we employed a MAML initialization method. This approach provides optimal initial policies for the ECS agents, significantly reducing learning time and enabling rapid adaptation to changing network conditions. The integration of MADDPG and SAC establishes a powerful dual-layer optimization framework, wherein MADDPG efficiently manages global resource slicing across the network, while SAC ensures real-time adaptability and fine-grained resource allocation at the network edge. This synergistic approach delivers superior scalability and efficiency, outperforming existing methods in managing network slices under dynamic conditions. Thus, the proposed MAARS algorithm offers a robust and effective solution for optimizing resource allocation and network performance in complex MEC environments. Its adaptability to varying network conditions makes it a promising approach for managing the evolving demands of next-generation networks. Future research should focus on further enhancing the performance of MAARS and extending its applicability to more complex and heterogeneous network scenarios.

## Figures and Tables

**Figure 1 sensors-24-07760-f001:**
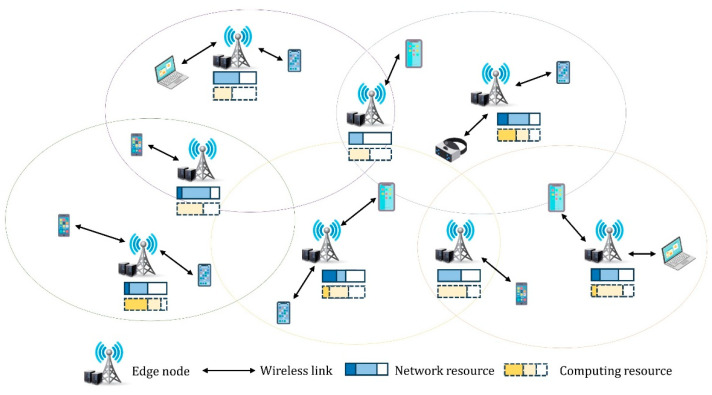
System architecture.

**Figure 2 sensors-24-07760-f002:**
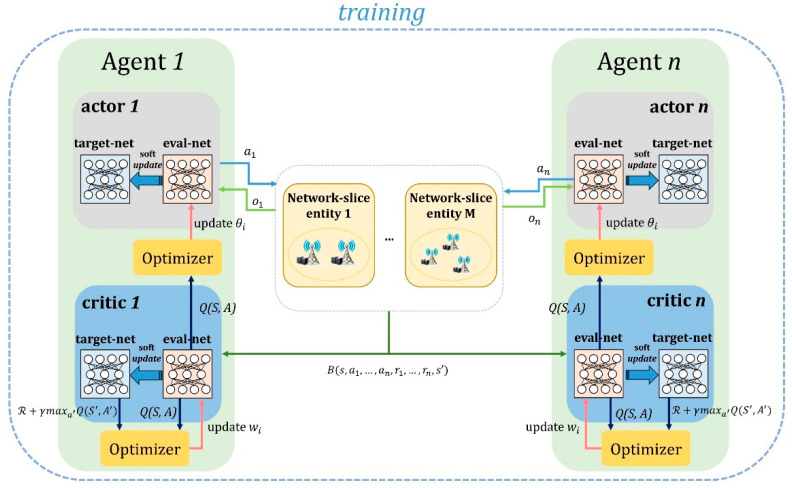
Illustration of the MADDPG-based ECS algorithm.

**Figure 3 sensors-24-07760-f003:**
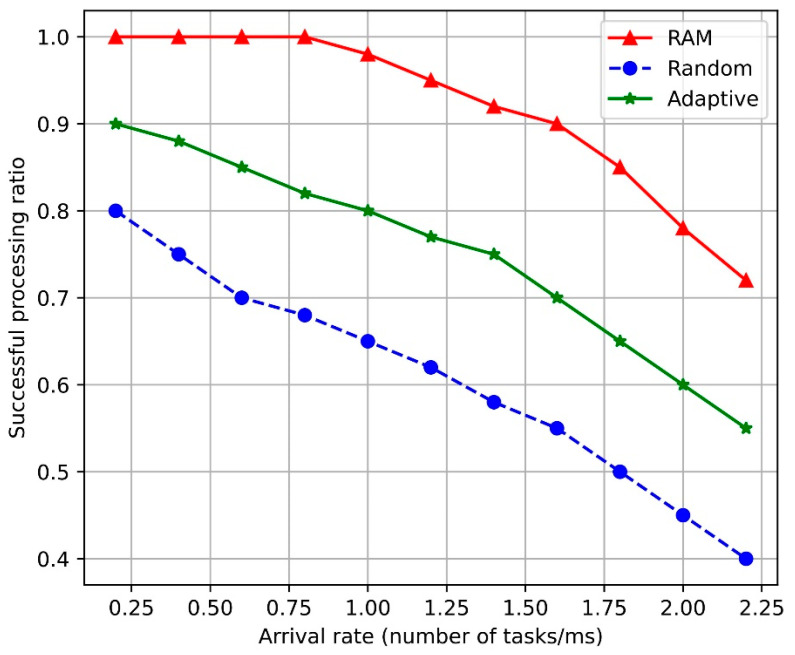
Task-completion ratio vs. arrival rate. RAM: resource-allocation management.

**Figure 4 sensors-24-07760-f004:**
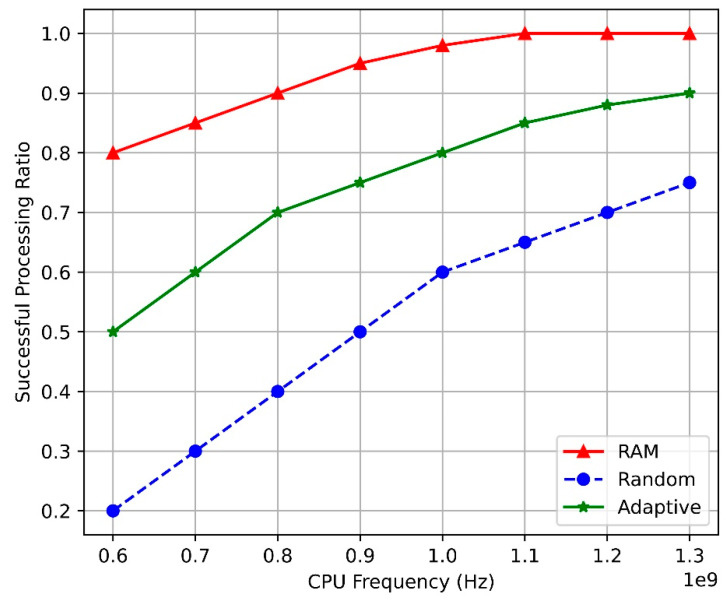
Task-completion ratio vs. CPU frequency.

**Figure 5 sensors-24-07760-f005:**
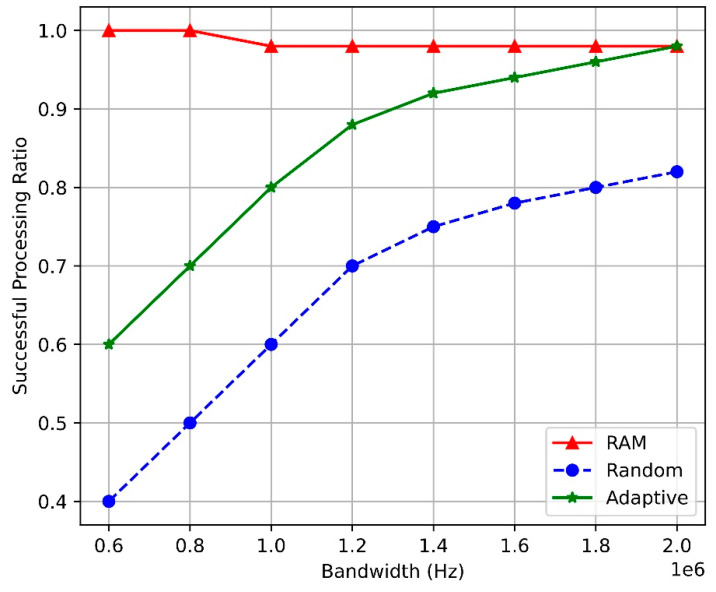
Task-completion ratio vs. bandwidth.

**Figure 6 sensors-24-07760-f006:**
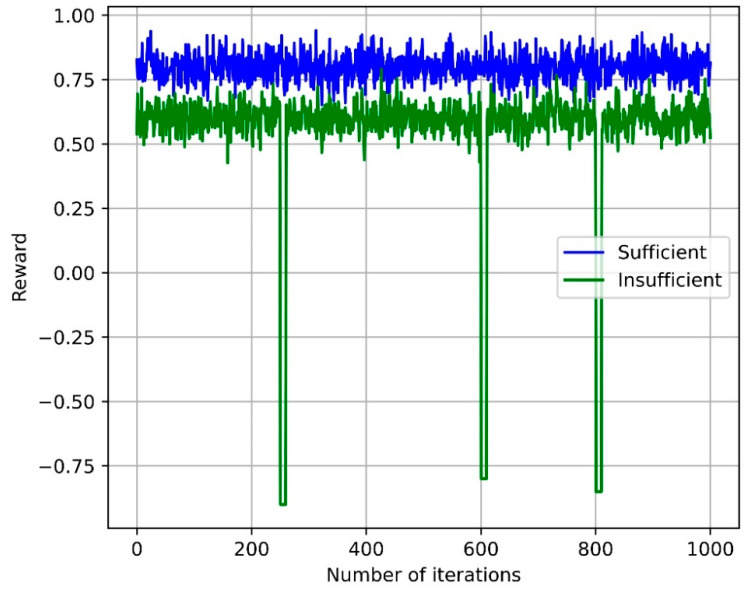
Reward values vs. number of iterations.

**Figure 7 sensors-24-07760-f007:**
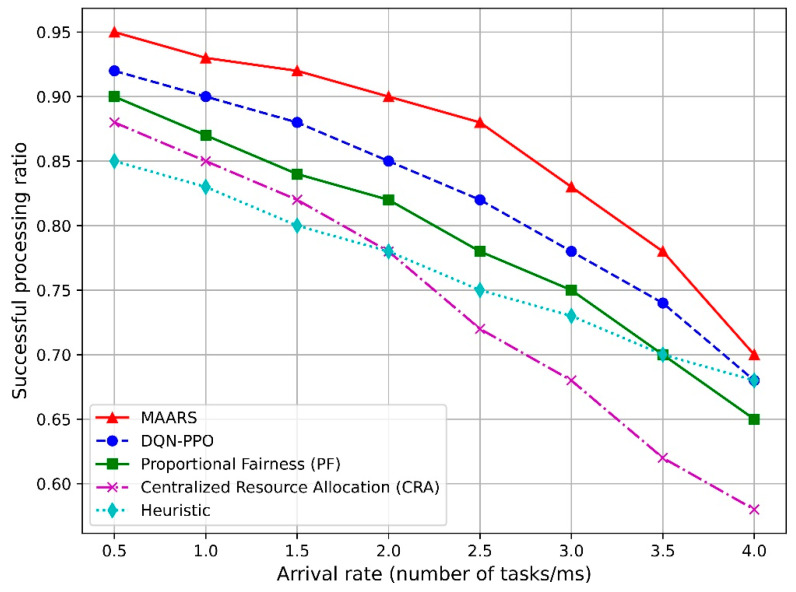
Task-completion ratio vs. arrival rate.

**Figure 8 sensors-24-07760-f008:**
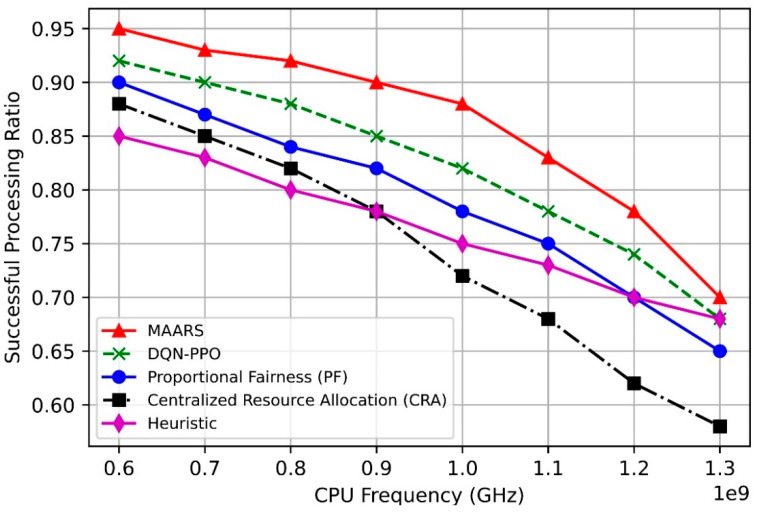
Task-completion ratio vs. CPU frequency.

**Figure 9 sensors-24-07760-f009:**
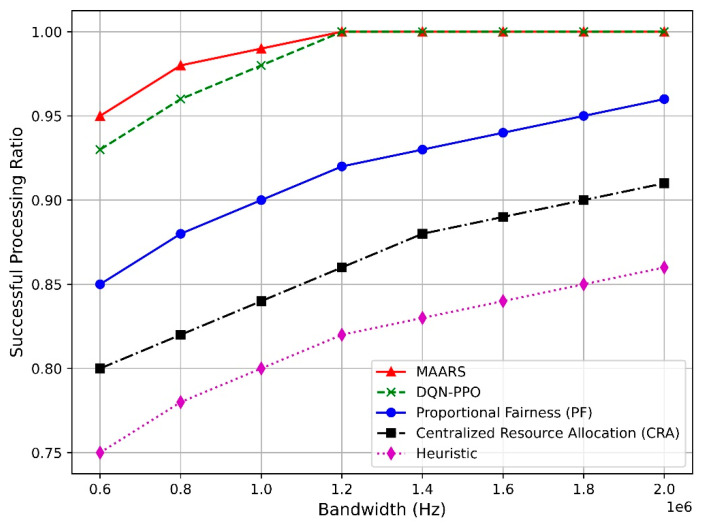
Task-completion ratio vs. bandwidth.

**Figure 10 sensors-24-07760-f010:**
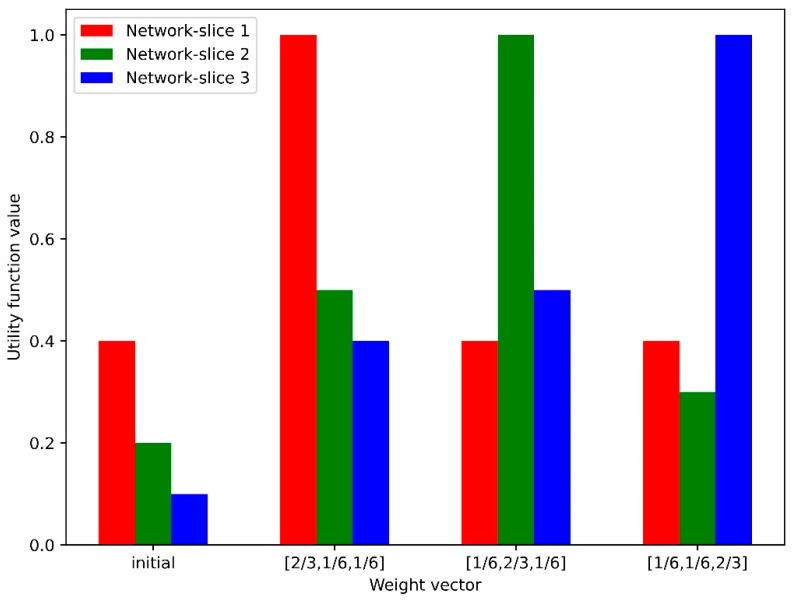
Utility-function values vs. weight vectors.

**Figure 11 sensors-24-07760-f011:**
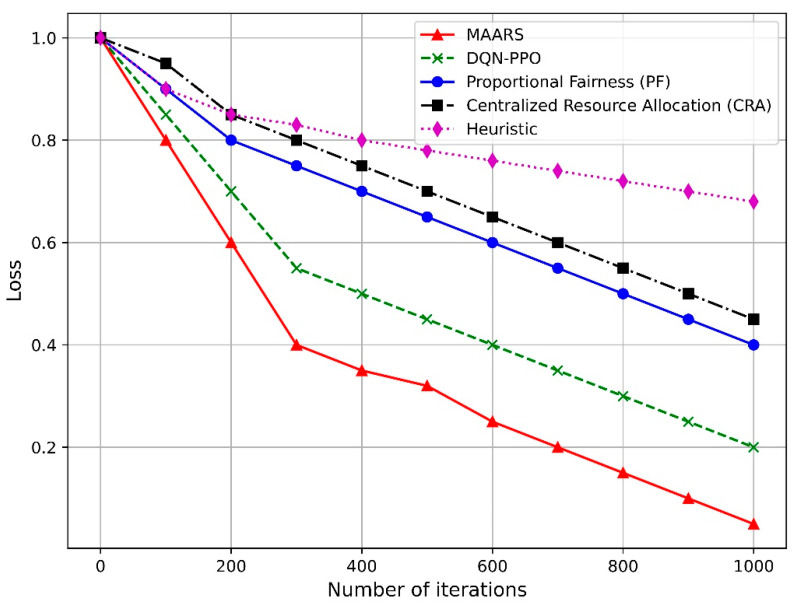
Loss ratio vs. number of iterations.

**Table 1 sensors-24-07760-t001:** Mathematical notations of the system model.

Notation	Definition
$R_{m}(t)$	Resource-allocation set for *m*-th logical network at time *t*
$B_{m}(t)$	Allocated bandwidth for the *m*-th network at time *t*
$F_{m}(t)$	Allocated computing resources for the *m*-th network at time *t*
$C_{m}(t)$	Allocated caching resources for the *m*-th network at time *t*
$X_{m}$	Resource-entity set for the *m*-th logical network
$W_{w,\;m}(t)$	Allocated resource entity for the *w*-th service in the *m*-th network
${Svr}_{k,\;m}(t)$	Server-selection indicator for task allocation
$r_{k}(t)$	Data-transmission rate for task *k* at time *t*
$\Gamma_{k,\;trans}\left(t\right)$	Transmission delay for task *k* at time *t*
$\Gamma_{k,\;proc}\left(t\right)$	Processing delay for task *k* at time *t*
$\Gamma_{k,\;total}\left(t\right)$	Total delay for task *k* at time *t*
$\rho_{w,m,n}\left(t\right)$	Completion ratio for service *w* in the *m*-th network slice at time *t*
$\rho_{m,total}\left(t\right)$	Total task-completion ratio for the *m*-th network slice at time *t*
$\rho_{avg}\left(t\right)$	Weighted average of task-processing ratio at time *t*
$B_{w,\;m,\;total}(t)$	Total bandwidth allocated for *w*-th service in the *m*-th network at time *t*
$F_{w,\;m,\;total}(t)$	Total computing resources allocated for the *w*-th service in the *m*-th network at time *t*
$B_{n}(t)$	Allocated bandwidth for the *n*-th network entity at time t
$F_{n}(t)$	Allocated computing resources for the *n*-th network entity at time *t*
$B_{network,\;max}(t)$	Maximum network bandwidth
$F_{network,\;max}(t)$	Maximum computing resources in the network
πECS	Resource-allocation policy for enhanced cognitive scheduling
πARS	Resource-allocation policy for adaptive resource scheduling
$L\left(\theta_{i}\right)$	Loss function for agent *i*
$y_{i}$	Target learning *Q*-value for agent *i*
$o_{m}\left(t\right)$	Local observation vector for the *m*-th agent at time *t*
*s(t)*	State vector for all agents at time *t*
$a_{m}\left(t\right)$	Action vector for the *m*-th agent at time *t*
$a\left(t\right)$	Joint-action vector for all agents at time *t*
$r_{m}\left(t\right)$	Reward function for agent m at time *t*
$\phi_{i}^{\prime }$	Updated policy parameters for agent *i*
$\phi_{i}^{*}$	Optimal policy parameters for agent *i*
Θ	Meta-policy parameters
$\Theta_{new}^{\prime }$	Updated meta-policy parameters
*State(t)*	State vector representing the network condition at time *t*
*Action(t)*	Action vector representing the resource allocation at time
*reward(t)*	Reward function to optimize resource-usage at time *t*
$\vartheta_{max}$	Maximum allowable delay
$\vartheta_{actual,\;i}(t)$	Actual delay for user *i* at time *t*

**Table 2 sensors-24-07760-t002:** Hyperparameters employed in the simulations.

Parameter	Value
Number of ENs (N)	10
Number of network-slice entities (M)	4
Transmission power $p_{k}(t)$	23 dBM
Noise power $\left(N_{0}\right)$	−95 dBm
CPU frequency	2.5 GHz
Bandwidth	5 MHz
Batch size	64
Learning rate (α)	0.001
Discount factor (γ)	0.99
Experience pool size	1000

## Data Availability

The data that support the findings of this study are available upon request from the corresponding author, I.J. The data are not publicly available because they contain information that can compromise the privacy of the research participants.

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
