# Peer review of "MAARS: Multiagent Actor–Critic Approach for Resource Allocation and Network Slicing in Multiaccess Edge Computing"

_sensors, 2024, doi:10.3390/s24237760_

Round 1

Reviewer 1 Report

Comments and Suggestions for Authors

This paper presents a new algorithm, MAARS, based on multi-agent reinforcement learning to address resource allocation and network slicing issues in multi-access edge computing (MEC). The algorithm divides the problem into two subproblems: core-to-edge slicing (ECS) and autonomous resource slicing (ARS) to efficiently schedule user computing tasks. MAARS combines multi-agent deep deterministic policy gradient (MADDPG) and soft actor-critic (SAC) techniques, and simulation results show its superiority over baseline algorithms in terms of task completion rate.Overall, the paper makes valuable contributions to resource optimization and network slicing in MEC networks but lacks some clarity in innovation and has deficiencies in experimental design.

 At the same time, there are some issues that need to be addressed:

1. The integration of MADDPG and SAC is not sufficiently explained in terms of innovation, and the paper lacks details on how it improves upon existing methods.

2. The paper lacks details about the role of model-agnostic meta-learning (MAML) in improving algorithm performance.

3. In Section 2, the descriptions of existing methods and the innovations of the MAARS algorithm are overly detailed and somewhat redundant.

Therefore, it is recommended to revise and resubmit.

Comments on the Quality of English Language

The English language in the paper is generally clear and understandable, but there are some areas where clarity and precision could be improved. A few sentences are slightly lengthy or complex, which could be simplified for better readability. Additionally, attention to consistent technical terminology and grammatical accuracy would enhance the overall quality. Overall, the language is of a good standard, but minor revisions would improve its flow and clarity.

Author Response

Comments 1. The integration of MADDPG and SAC is not sufficiently explained in terms of innovation, and the paper lacks details on how it improves upon existing methods.

Response: Thank you for your valuable comments. To clarify the innovative aspects of integrating MADDPG and SAC, we have revised the relevant sections of the manuscript. First, we have clearly explained the originality of the proposed approach through the integration of MADDPG and SAC and added specific cases and discussions on how this integration complements the limitations of existing methods (Page 3, Lines 78–103), (Page 10, Lines 343–344), (Page 11, Lines 405–411), (Page 15, Lines 558–564), (Page 26, Lines 823–831), (Page 27, Lines 840–845). Additionally, we have included additional experimental results and analyses to demonstrate the performance improvement (Page X, Lines XX–XX). We believe that these changes clearly highlight the strengths of the proposed method compared with existing methods.

Comments 2. The paper lacks details about the role of model-agnostic meta-learning (MAML) in improving algorithm performance.

Response: We would like to thank the reviewer for their insightful feedback on the role of MAML. Accordingly, we have revised the manuscript to include a more detailed explanation of MAML’s contribution to the proposed framework. Specifically, in the Introduction (Page 2, Lines 70–77), we have elaborated on how MAML improves the learning efficiency and adaptability of the MADDPG algorithm by providing optimized meta-policy parameters. This leads to faster convergence and improved stability under dynamic network conditions. In Section 4.3 (Page 13, Lines 477–496), we have provided a detailed description of the MAML-based initialization algorithm, clarifying the role of the inner and outer loops and how the meta-policy parameters are pre-trained and fine-tuned for the ECS task. We have also included a detailed analysis of the impact of MAML on performance in Section 4.4 (Page 15, Lines 548–564), focusing on its role in reducing convergence time and improving resource allocation adaptability under volatile conditions. The Conclusion (Page 27, Lines 833–852) has also been updated to emphasize the essential contributions of MAML to resource optimization and robust adaptation in MEC networks.

Comments 3. In Section 2, the descriptions of existing methods and the innovations of the MAARS algorithm are overly detailed and somewhat redundant.

Response: Thank you for your constructive feedback. Accordingly, we have revised Section 2 (Related Work) to streamline the descriptions of existing methods and the innovations of the MAARS algorithm. Additionally, we have condensed the descriptions of methods while retaining their significant contributions and limitations (Page 3, Lines 110–152). Furthermore, we have removed redundant phrases and revised the paragraph describing the MAARS algorithm (Page 18, Lines 653–665) to clearly distinguish it from previous solutions and focus on its novel features and advantages.

Comments on the Quality of English Language

The English language in the paper is generally clear and understandable, but there are some areas where clarity and precision could be improved. A few sentences are slightly lengthy or complex, which could be simplified for better readability. Additionally, attention to consistent technical terminology and grammatical accuracy would enhance the overall quality. Overall, the language is of a good standard, but minor revisions would improve its flow and clarity.

Response: We appreciate the reviewer's feedback on the language and clarity of the manuscript. We have carefully reviewed the entire manuscript to identify and revise any areas where clarity and precision could be improved. Specifically, we have focused on:

  • Simplifying lengthy or complex sentences: We have broken down long sentences into shorter, more manageable ones to enhance readability.
  • Ensuring consistent technical terminology: We have reviewed the manuscript to ensure that technical terms are used consistently throughout.
  • Improving grammatical accuracy: We have paid close attention to grammar, punctuation, and syntax to eliminate any errors.
  • Enhancing flow and clarity: We have made revisions to improve the overall flow and clarity of the writing.

We believe these revisions have significantly improved the readability and clarity of the manuscript.

Reviewer 2 Report

Comments and Suggestions for Authors

In this article, the authors present an algorithm to solve resource-allocation and network-slicing problems in multiaccess edge computing (MEC) networks. This study makes a significant contribution to enhancing resource-optimization and network-slicing efficiency in MEC networks. The presented spectrum sensing technique is interesting however, I do not recommend its acceptance in the present form. I have following concerns.

1)                   In the presented resource allocation Algorithms, I did not find about the authors concern about spectrum as well as power allocation. Power allocation in the communication networks is directly concern about the interference management which is challenging task in high traffic environment. I hope the following article may assist to authors regarding these issues.

Ngene, C. E., Thakur, P., & Singh, G. (2023). Power allocation strategies for 6G communication in VL-NOMA systems: an overview. Smart Science11(3), 475-518.

2)                   Complexity of Algorithm 1, Algorithm 2 and Algorithm 3 need to discuss in detail particularly space and time complexity.

3)                   How does the authors validate the proposed algorithm?      

Author Response

Comments 1)  In the presented resource allocation Algorithms, I did not find about the authors concern about spectrum as well as power allocation. Power allocation in the communication networks is directly concern about the interference management which is challenging task in high traffic environment. I hope the following article may assist to authors regarding these issues.

[11] Ngene, Thakur, & Singh (2023). Power allocation strategies for 6G communication in VL-NOMA systems: an overview. Smart Science11(3), 475-518.

- We appreciate your valuable feedback regarding the inclusion of spectrum and power allocation in the resource allocation algorithm. Accordingly, we have revised the manuscript to address these aspects. In the Introduction section (Page 2, Lines 60–63), we have included a discussion on the importance of spectrum and power allocation in MEC networks, emphasizing their role in optimizing resource management and mitigating interference in high-traffic environments. We have also referenced the suggested article by Ngene et al. (2023) to support this discussion.

Comments 2) Complexity of Algorithm 1, Algorithm 2 and Algorithm 3 need to discuss in detail particularly space and time complexity.

- We appreciate your insightful feedback on discussing the computational complexity of the proposed algorithms. Accordingly, we have added a new section, "4.5. Computational Complexity" (Page 17, Lines 621–638), which provides a detailed analysis of the time and space complexities of Algorithms 1, 2, and 3. For Algorithm 1, we have explained how the computational effort scales with the number of tasks and resources, while the state of the functions primarily determines the memory usage. Next, Algorithm 2 involves matrix operations to optimize resource allocation, and the task-resource data and optimization parameters affect the processing time and memory usage. Finally, Algorithm 3 uses reinforcement learning to update the policy, and the computational demand varies with the number of slices and iterations, while the memory usage reflects the storage of policy parameters. Combining these is ensures that the framework maintains the desired efficiency and scalability.

Comments 3) How does the authors validate the proposed algorithm?      

- Thank you for your inquiry about the validation of the proposed algorithm. We validated the algorithm through extensive simulations under realistic MEC network conditions. The simulation environment was constructed using Python and PyTorch, and the experiments were conducted on a system comprising an Intel Core i7 CPU and an NVIDIA RTX GPU. The performance of the proposed algorithm was compared with that of several baseline methods, including heuristic-based, DQN-based, and A2C-based algorithms. The evaluation metrics included task success rate, resource utilization efficiency, and convergence speed during training. The simulation scenarios were designed to reflect various network conditions, including high traffic loads, varying user demands, and different resource availability. User task arrival was modeled as a Poisson process. Each experiment was repeated multiple times to ensure the reliability of the results. The results showed that the proposed algorithm outperforms the baseline methods, achieving a higher task completion rate, better resource utilization, and faster convergence, thereby validating its efficiency in managing resource allocation and network slicing in MEC networks. This information has been added to the revised manuscript (Page 17, Lines 639–670).

Round 2

Reviewer 2 Report

Comments and Suggestions for Authors

In the revised manuscript, the authors have incorporated almost all the comments and suggestions raised by the reviewers. Therefore, I recommend the acceptance of this manuscript for publications.